# Learning induces the translin/trax RNase complex to express activin receptors for persistent memory

Alan Jung Park[1‡], Robbert Havekes[1†§], Xiuping Fu[2†], Rolf Hansen[1†], Jennifer C Tudor[1†#], Lucia Peixoto[1¶], Zhi Li[2**], Yen-Ching Wu[2], Shane G Poplawski[1], Jay M Baraban[2], Ted Abel[1,3††*]

[1]Department of Biology, University of Pennsylvania, Philadelphia, United States; [2]Solomon H. Snyder Department of Neuroscience, Johns Hopkins School of Medicine, Baltimore, United States; [3]Molecular Physiology and Biophysics, Iowa Neuroscience Institute, Carver College of Medicine, University of Iowa, Iowa City, Iowa, United States

*For correspondence: ted-abel@uiowa.edu

[†]These authors contributed equally to this work

Present address: [‡]Department of Psychiatry, Columbia University, New York, United States; [§]Groningen Institute for Evolutionary Life Sciences, University of Groningen, Groningen, Netherlands; [#]Department of Biology, Saint Joseph's University, Philadelphia, United States; [¶]Washington State University Spokane, Washington, United States; [**]Department of Orthopaedic Surgery, Johns Hopkins School of Medicine, Baltimore, United States; [††]Iowa Neuroscience Institute, Departments of Molecular Physiology and Biophysics, Psychiatry and Biochemistry, University of Iowa Carver College of Medicine, Iowa City, United States

Competing interests: The authors declare that no competing interests exist.

**Abstract** Long-lasting forms of synaptic plasticity and memory require de novo protein synthesis. Yet, how learning triggers this process to form memory is unclear. Translin/trax is a candidate to drive this learning-induced memory mechanism by suppressing microRNA-mediated translational silencing at activated synapses. We find that mice lacking translin/trax display defects in synaptic tagging, which requires protein synthesis at activated synapses, and long-term memory. Hippocampal samples harvested from these mice following learning show increases in several disease-related microRNAs targeting the activin A receptor type 1C (ACVR1C), a component of the transforming growth factor-β receptor superfamily. Furthermore, the absence of translin/trax abolishes synaptic upregulation of ACVR1C protein after learning. Finally, synaptic tagging and long-term memory deficits in mice lacking translin/trax are mimicked by ACVR1C inhibition. Thus, we define a new memory mechanism by which learning reverses microRNA-mediated silencing of the novel plasticity protein ACVR1C via translin/trax.

DOI: https://doi.org/10.7554/eLife.27872.001

## Introduction

The synthesis of plasticity proteins at activated synapses is critical for persistent synaptic plasticity and long-term memory (*Doyle and Kiebler, 2011*; *Mayford et al., 2012*; *Redondo and Morris, 2011*). Deficits in this process are thought to play a prominent role in neurodevelopmental and psychiatric disorders (*Liu-Yesucevitz et al., 2011*). Therefore, identification of the molecular pathways linking synaptic activation to local translation of key plasticity proteins is a top priority as it promises to yield valuable insights into the etiology of and treatment for these debilitating disorders.

Neuronal stimulation can trigger rapid translation of plasticity transcripts by reversing microRNA-mediated translational silencing. For example, pharmacological activation of cultured neurons (*Huang et al., 2012*) and electrical stimulation of the perforant pathway of rodents in vivo (*Joilin et al., 2014*) elicit rapid suppression of microRNA-mediated translational silencing. In the amygdala, fear conditioning induces microRNA level changes to form fear memory (*Griggs et al., 2013*). Thus, synaptic stimuli and learning may drive the translation of plasticity transcripts by reversing microRNA-mediated silencing. However, the mechanism linking synaptic stimulation or learning to microRNA-mediated expression of key plasticity proteins remains elusive.

One candidate for mediating this mechanism is a protein complex composed of translin and its partner protein translin-associated factor X (trax), a catalytic subunit with endoRNase activity

(*Finkenstadt et al., 2000*; *Liu et al., 2009*; *Tian et al., 2011*; *Ye et al., 2011*). In this complex, two homomeric translin dimers and two heteromeric translin/trax dimers form a hetero-octomer (*Tian et al., 2011*; *Ye et al., 2011*). Translin/trax is brain enriched (*Han et al., 1995a*) and translocates to dendrites in response to neuronal stimulation (*Han et al., 1995b*; *Wu et al., 2011*). Initially, the RNase activity of translin/trax was not appreciated and, hence, it was thought to function as an RNA-binding protein complex involved in RNA trafficking. However, following recognition of its catalytic activity, recent studies have revealed that transln/trax suppresses microRNA-mediated translational silencing by degrading microRNAs in non-neuronal cells (*Asada et al., 2014*). Therefore, translin/trax is ideally positioned to link synaptic activity to the synthesis of plasticity proteins at activated synapses.

To evaluate the role of translin/trax in synaptic plasticity and memory formation, we utilized translin knockout (KO) mice in this study. As trax protein is unstable in the absence of translin, translin KO mice are completely devoid of both proteins (*Chennathukuzhi et al., 2003*; *Yang et al., 2004*). Previous behavioral characterization of these mice has revealed that they have intact long-term spatial and contextual memory assessed by the water maze and fear conditioning (*Stein et al., 2006*). However, behaviors from such aversive tasks need careful interpretation as translin KO mice display reduced anxiety-related behavior in addition to developmental adaptations (*Chennathukuzhi et al., 2003*; *Stein et al., 2006*). Therefore, designing experiments to assess memory formation in translin KO mice should account for their developmental as well as baseline alterations.

Prior studies have not examined the synaptic function of translin/trax. To investigate the role of translin/trax in synaptic control of plasticity-related protein synthesis, we employed a synaptic tagging paradigm in which transient potentiation in one pathway becomes persistent in response to strong stimulation in a separate pathway (*Frey and Morris, 1997*). This form of heterosynaptic plasticity is ideal to evaluate the synaptic function of translin/trax as it requires the synthesis of plasticity proteins at synapses that are tagged by synaptic stimulation (*Redondo and Morris, 2011*). Moreover, in addition to examining heterosynaptic interactions between the two pathways, this synaptic tagging paradigm also allows us to assess conventional homosynaptic plasticity in each pathway. Because the identity of key plasticity proteins critical for synaptic tagging remains elusive, establishing the mechanism by which translin/trax mediates the synthesis of key plasticity proteins required for persistent synaptic plasticity will significantly advance our knowledge of this important process.

Here, given that translin KO mice display normal behavior in the open field (*Stein et al., 2006*), we investigated the role of the translin/trax in long-term memory using an open field-based object-location task. Also, viral reinstatement of translin/trax in the hippocampus of adult translin KO mice confirmed that the observed phenotypes were not due to non-specific effects of constitutive deletion of translin/trax. This 'rescue' experiment further allows us to conclude that translin/trax acts within the hippocampus to mediate synaptic tagging and long-term object memory. Finally, the search for targets of the microRNAs regulated by translin/trax after learning identified the novel plasticity protein ACVR1C, which is critical for persistent synaptic plasticity and long-term memory formation.

## Results

### Translin/trax is required for synaptic tagging and long-term object location memory

We first investigated whether the lack of translin/trax impacts synaptic plasticity using a synaptic tagging paradigm. Two independent pathways (S1 and S2) in hippocampal area CA1 received 4-train and 1-train stimulation, respectively, separated by 30 min (*Figure 1A*). Hippocampal slices from translin KO mice that lack translin/trax showed similar 4-train-induced long-lasting potentiation to wildtype (WT) slices in pathway S1 (*Figure 1B1*). In pathway S2, however, 1-train stimulation following S1 stimulation failed to induce persistent potentiation in translin KO slices (*Figure 1B2*). Importantly, translin KO mice did not show altered 1-train-induced short-lived potentiation when administered alone (*Figure 1—figure supplement 1A*). Basal synaptic transmission measured by the input-output relationship and PPF was also unaltered in translin KO mice (*Figure 1—figure supplement 1B*; KO: $5 \pm 1.9$, n = 6; WT: $3.8 \pm 0.6$, t-test, p=0.6, and *Figure 1—figure supplement 1C*; two-way repeated measures ANOVA, $F_{(1,10)} = 0.2$, p=0.8).

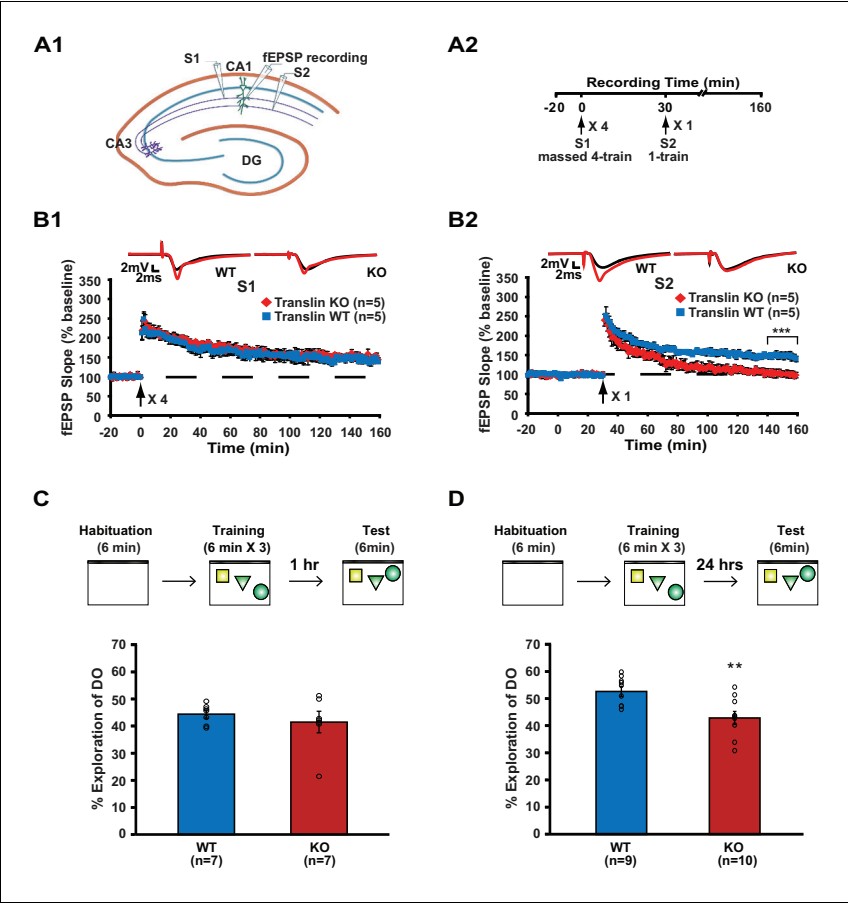

**Figure 1.** Translin KO mice display impaired synaptic tagging and long-term memory formation. (**A1**) A schematic diagram showing the two independent inputs (S1 and S2) converging onto postsynaptic neurons in CA1. (**A2**) Experimental scheme. (**B**) Slices from translin KO mice showed unaltered 4-train long-lasting potentiation in pathway S1 (B1; two-way repeated-measures ANOVA, $F_{(1,8)}$ = 0.03, p=0.9), but displayed reduced persistent potentiation after 1-train stimulation in pathway S2 (**B2**); two-way repeated-measures ANOVA, $F_{(1,8)}$ = 36.5, p=0.0003. Average traces for the baseline (black) and the last 20 min (red) are shown above each graph. (**C and D**) Top, experimental scheme. Translin KO mice explored the displaced object (DO) significantly less than WT littermates 24 hr after training (**C**; t-test, p=0.004), while explored DO similar to WT littermates 1 hr after training (**D**; t-test, p=0.5). ** indicates p<0.005. *** indicates p<0.0005. n, number of mice. Data are represented as mean ±SEM.

DOI: https://doi.org/10.7554/eLife.27872.002

The following figure supplements are available for figure 1:

**Figure supplement 1.** Short-lived LTP and basal synaptic transmission are unaltered in translin KO mice.

DOI: https://doi.org/10.7554/eLife.27872.003

**Figure supplement 2.** Exploratory behavior during object-location memory task is unaltered in translin KO mice.

DOI: https://doi.org/10.7554/eLife.27872.004

Because heterosynaptic integration of synaptic inputs during synaptic tagging provides a cellular model of the associative memory process, we next examined the role of translin/trax in memory formation. Associative memory can be investigated using a hippocampus-dependent long-term object-location memory task, in which mice learn and remember the location of each object relative to external cues or other objects (*Oliveira et al., 2010*). If the spatial location of each object is properly encoded during training, mice identify the displaced object during testing 24 hr after training, showing increased preference of what is perceived as a spatially novel stimulus. This task involves initial exposure to an open field (context) and then subsequent exposure to objects (*Figure 1C and D*), experiences that potentially mirror the independent synaptic inputs that are integrated in synaptic

tagging. Mice were trained and the preference for the displaced object 24 hr after training was compared between WT and translin KO mice. Compared to WT littermates, preference for the displaced object was significantly reduced in translin KO mice (*Figure 1D*; WT: 52.3 ± 1.8%, n = 9; KO: 42.6 ± 2.4%). However, 1 hr short-term memory was unaffected in translin KO mice (*Figure 1C*; WT: 44.4 ± 1.5%; KO: 41.5 ± 4%), suggesting that translin KO mice are able to acquire relevant information, but have deficits in memory consolidation. In line with previous findings that translin KO mice exhibit unaltered exploratory behavior in an open field (*Stein et al., 2006*), the total object exploration time during training and testing did not differ across groups (*Figure 1—figure supplement 2*). Overall, these data indicate that translin/trax is critical for heterosynaptic strengthening of short-lived potentiation and long-term associative memory formation.

## Loss of translin/trax alters microRNA levels after learning

We hypothesized that dysregulation of microRNA-mediated translational silencing underlies the defects in synaptic tagging and object location memory displayed by translin KO mice. Thus, we first examined whether translin/trax co-localizes with P-bodies, organelles enriched with components of the RNA-induced silencing complex (RISC) (*Eulalio et al., 2007*). Immunostaining of primary hippocampal neurons revealed that nearly all identified puncta for trax, the catalytic subunit of translin/trax (*Liu et al., 2009*), co-localized with puncta for GW182, a P-body marker, in dendrites. Conversely, many GW182-positive puncta were not trax-positive. Together with previous findings that translin/trax degrades microRNAs, these data indicate that translin/trax is well-positioned to regulate microRNA-mediated translational silencing (*Figure 2*).

Translin deletion selectively impairs long-term object location memory and synaptic tagging without impairing baseline behavior and synaptic properties or the induction of long-lasting potentiation by high frequency stimulation. This suggests that translin/trax may elicit degradation of microRNAs in response to specific stimuli. To define learning-induced targets of translin/trax, we determined whether translin KO mice display altered hippocampal microRNA levels after object location memory training. A microRNA PCR array (MIMM-107ZE-1, Qiagen), which includes 84 microRNAs implicated in nervous system development or disease, was used to probe for dysregulated microRNAs, and the candidate microRNAs were verified with real-time RT-PCR (see Methods). Compared to WT littermates, translin KO mice show significant increases in the levels of let-7c-5p, miR-125b-5p, miR-128–3p, and miR-9–3p in the hippocampus 30 min following training. Surprisingly, however, there was no difference across candidate microRNAs between translin KO mice and WT littermates kept under homecage conditions (*Figure 3A*). These microRNAs are brain enriched and implicated in neuropsychiatric disorders, including schizophrenia, Alzheimer's Disease and anxiety (*Edbauer et al., 2010*; *Lukiw, 2007*; *Muiños-Gimeno et al., 2009*). In contrast, miR-409–3p was not a learning specific target of translin/trax as its levels in the hippocampus were increased in translin KO mice both after training and in home cage conditions (p=0.00009 and p=0.03, respectively; data not shown). These data suggest that translin/trax mediates rapid degradation of selected microRNAs after learning in vivo, and could thereby reverse translational silencing of plasticity transcripts targeted by these microRNAs.

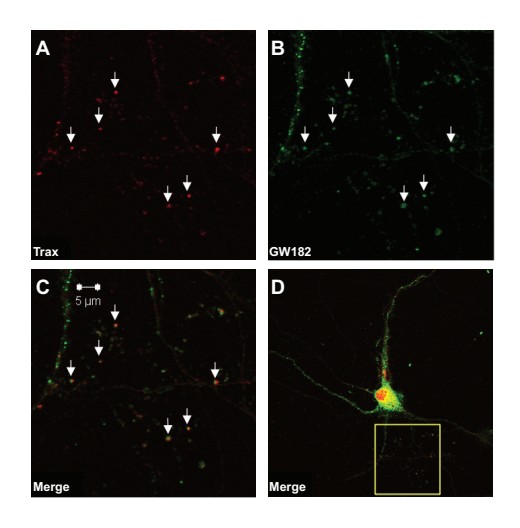

**Figure 2.** Trax co-localizes with P-bodies. Immunostaining images display a rat hippocampal neuron stained for trax (**A** red) and GW182, a marker of P-bodies (**B** green). (C) Merged image. (D) The box indicates the area depicted at higher magnification in the other three panels. Arrowheads indicate puncta that are co-labelled by trax and GW182.
DOI: https://doi.org/10.7554/eLife.27872.005

## Translin/trax reverses microRNA silencing of ACVR1C transcript after learning

To identify transcripts that undergo training-induced reversal of silencing in a translin/trax dependent fashion, we searched for target genes of the microRNAs elevated in translin KO mice following training (see Methods). To narrow our search, we focused on microRNA targets conserved between human and mouse. As miR-9–3p did not have any target transcripts that meet this criterion, we restricted our analysis to targets of let-7c-5p, miR-125b-5p and miR-128–3p (*Figure 3B*, *Figure 3— source data 2*). Of the thirteen transcripts targeted by all three of these microRNAs, we chose Activin receptor 1C (ACVR1C, or activin-like kinase 7, ALK7) and vang-like 2 (VANGL2) as leading candidates because they have been implicated in hippocampal plasticity (*Inokuchi et al., 1996*; *Nagaoka et al., 2014*). ACVR1C was of particular interest because it has the highest aggregate context score (*Grimson et al., 2007*) within this group (*Figure 3—source data 1*). Although CPEB3 has a role in memory formation, it was excluded as a likely candidate because the electrophysiological phenotypes displayed by adult mice lacking CPEB3 do not match those observed in translin KO mice (*Chao et al., 2013*; *Fioriti et al., 2015*).

To examine whether let-7c, miR-125b-5p and miR-128–3p mediate silencing of the ACVR1C transcript, we conducted luciferase reporter assays. Consistent with their predicted context scores (*Figure 3—source data 1*), a fragment of the ACVR1C 3'UTR containing target sites for let-7c-5p and miR-128–3p produced robust silencing of a luciferase reporter in response to treatment with either of these microRNAs, but not to miR-125–5p (*Figure 4A and B*). The luciferase activity of the positive control TRIM71, which has target sites for let-7c-5p and miR-125–5p, was suppressed by let-7c-5p and miR-125–5p, but not miR-128–3p (*Figure 4C*). The effects of these microRNAs were prevented by the treatment with their inhibitors (*Figure 4D*) or mutating their target sites in the 3'UTR (*Figure 4E*). These studies confirm that ACVR1C is subject to silencing directly by let-7c and miR-128–3p.

Because neuronal activity induces dendritic translocation of translin (*Wu et al., 2011*), we reasoned that learning may induce synaptic localization of translin, which subsequently regulates the expression of target genes. Indeed, translin protein levels increased in hippocampal synaptosomes 30 min after training (*Figure 5*; home cage: $100 \pm 7.3\%$, training: $142 \pm 10\%$). Moreover, training increased synaptosomal ACVR1C protein levels in WT littermates, but not in mice lacking translin/trax, although baseline ACVR1C levels were similar between the groups (*Figure 5*; $WT_{homecage}$: $100 \pm 8.7\%$, $WT_{training}$: $198.8 \pm 28.6\%$; $KO_{homecage}$: $102 \pm 18.4\%$, $KO_{training}$: $106.4 \pm 8.7\%$). Synaptosomal VANGL2 levels were unaffected by training or genotype (*Figure 5*). Thus, increased synaptic expression of ACVR1C protein following training requires translin/trax. Together with luciferase studies, these findings reveal that training induces synaptic ACVR1C expression by translin-mediated reversal of microRNA silencing of this transcript.

## ACVR1C is required for the maintenance of synaptic tagging and long-term memory

Because our data imply that memory formation requires translin/trax-mediated synaptic expression of ACVR1C within minutes following neuronal activity, we determined whether immediate blockade of ACVR1C function after synaptic stimulation mimics the defects in synaptic plasticity and object location memory observed in translin KO mice. We used a pharmacological inhibitor of ACVR1C, SB431542 (*Inman et al., 2002*), to investigate the requirement for ACVR1C in memory formation and synaptic plasticity with temporal precision. Because the lack of translin/trax specifically impairs heterosynaptic strengthening of pathway S2, SB431542 treatment was started after pathway S1 was potentiated. We found that SB431542 did not alter long-lasting potentiation in pathway S1 (*Figure 6A1*), but blocked heterosynaptic strengthening of pathway S2 (*Figure 6A2*) in hippocampal slices from WT mice. Consistent with previous findings that activin signaling is not required for basal synaptic transmission (*Ageta et al., 2010*), SB431542 treatment did not alter baseline synaptic responses in pathway S2 (*Figure 6A2*). Moreover, treatment with SB431542 during the last 1 hr of the recordings impaired the maintenance phase of S2 potentiation (*Figure 6A3 and A4*). Thus, these findings indicate that ACVR1C is required for persistent heterosynaptic strengthening of pathway S2 and is involved in post-tagging maintenance of this form of heterosynaptic plasticity. Behaviorally, intrahippocampal infusion of SB431542 immediately after training impaired long-term object-

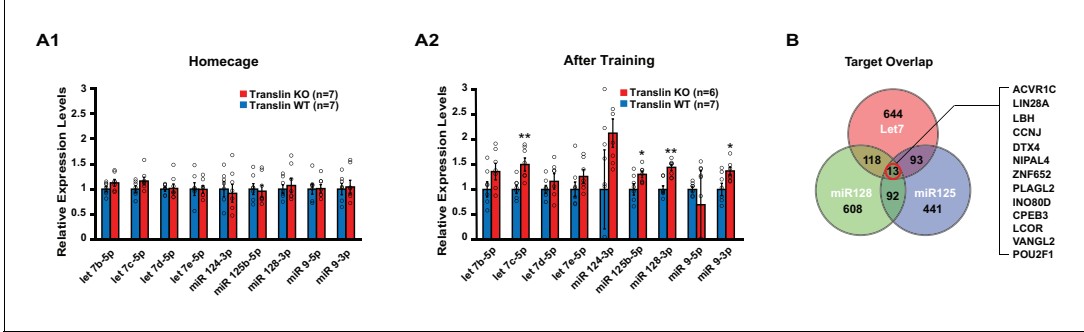

**Figure 3.** Translin KO mice display increased levels of several microRNAs targeting the activin A receptor type 1C (ACVR1C) 30 min following training in the object location memory task. Real-time RT-PCR was performed on hippocampal samples to validate candidate microRNAs. (**A**) Left, no significant difference was observed between translin KO and WT hippocampus in homecage condition. Right, the levels of let-7c-5p (t-test, p=0.003), miR-125b-5p (t-test, p=0.04), miR-128–3p (t-test, p=0.001) and miR-9–3p (t-test, p=0.03) were increased 30 min after training in translin KO hippocampus relative to WT hippocampus. (**B**) Venn diagram shows 13 overlapping targets of Let-7, miR-128 and miR-125. The 13 target genes are listed in a descending order of prediction context scores. ACVR1C has the highest context score. * indicates p<0.05, ** indicates p<0.005. Data are represented as mean ±SEM.

DOI: https://doi.org/10.7554/eLife.27872.006

The following source data is available for figure 3:

**Source data 1.** Common target genes of the conserved mRNA targets of let-7, miR-125 and miR-128.
DOI: https://doi.org/10.7554/eLife.27872.007

**Source data 2.** Conserved mRNA targets of let-7 between humans and mice.
DOI: https://doi.org/10.7554/eLife.27872.008

location memory 24 hr after training in WT mice (**Figure 6B2**; vehicle: 52.1 ± 1.3%, SB 431542: 38.3 ± 1.3%). Intrahippocampal injection of SB431542 did not alter the total object exploration time during training (**Figure 6—figure supplement 1A**) and testing (**Figure 6—figure supplement 1B**; vehicle: 2.7 ± 0.3 s; SB431542: 2.9 ± 0.4 s). Finally, as our findings indicate that training-induced synaptic expression of ACVR1C is dependent on translin/trax, we examined the prediction that the inhibitory effect of SB431542 on object location memory should be occluded in translin KO mice. Even though the preference for the displaced object displayed by the translin KO mice was above chance level or 33% (p=0.0005), treatment with SB431542 had no further inhibitory effect (**Figure 6B2**; vehicle: 40.6 ± 2.5%; SB431542: 40.1 ± 2.1%). Together, these data indicate that the memory deficits observed in translin KO mice result from ACVR1C dysregulation.

## Restoration of translin/trax to physiological levels in hippocampal excitatory neurons rescues deficits in synaptic tagging and long-term memory

Our data suggest that long-term memory formation requires rapid stimulus-dependent action of translin/trax, which occurs within minutes after learning. To allow for rapid engagement of translin/trax after learning, we reinstated translin/trax to physiological levels in hippocampal excitatory neurons of adult translin KO mice using an adeno-associated virus (**Figure 7A and B**). The level of virally expressed translin reached WT levels 15 to 20 days after injection (**Figure 7C**; WT: 100 ± 15.9%, $KO_{15day}$: 105.6 ± 30.3%, $KO_{20day}$: 159.6 ± 24.4%). Viral translin expression also restored trax levels by 20 day post-injection (**Figure 7D**; WT littermates: 100 ± 14%, $KO_{20day}$: 93.9 ± 12.4%). Thus, we performed all subsequent viral experiments at this time point. Basal synaptic properties measured by the input-output relationship and PPF were not affected by viral expression of either translin or eGFP (**Figure 1—figure supplement 1B**; translin virus group: 3.6 ± 0.5, eGFP virus group: 3.5 ± 0.5, and **Figure 1—figure supplement 1C**; two-way repeated measures ANOVA, $F_{(1,10)}$ = 0.2, p=0.6). We found that the viral expression of translin in the hippocampus of adult translin KO mice rescued the deficits in synaptic tagging (**Figure 7E**).

Additionally, twenty days after injection, mice were trained and the preference for the displaced object 24 hr after training was compared between mice injected with either translin or eGFP virus and WT littermates not injected with virus (**Figure 7F**; WT: 51.7 ± 2.1%, $KO_{translin}$: 54.7 ± 2.4%, $KO_{eGFP}$: 41.9 ± 2%, $WT_{eGFP}$: 52.1 ± 2%). Compared to WT littermates, preference for the displaced

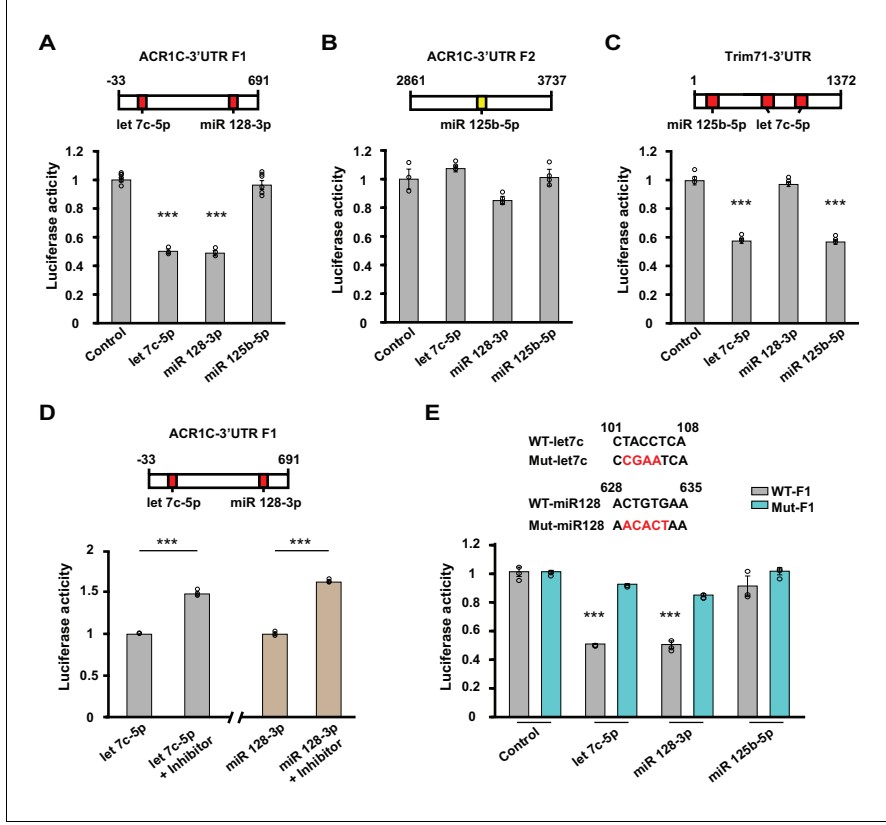

**Figure 4.** Let-7c-5p and miR-128–3p suppress luciferase activity of ACVR1C-3'UTR. (**A–D**) Top, schematic diagrams illustrating the location of target sites for selected microRNAs in 3'UTR fragments. Values shown above the boxes indicate nucleotide positions relative to the start of the 3'UTR. Red indicates target sites of given microRNAs with high context percentile scores (> 80th percentile), while yellow indicates a lower context score. Bottom, bar graphs illustrate normalized luciferase activity (relative to control) from cells transfected with the listed microRNA mimics at a concentration of 2 nM (n = 3–6 per group). (**A**) The luciferase activity of the F1 reporter construct was reduced by either let-7c-5p (target position: 101–108) or miR-128–3 p (target position: 628–635) treatment (one-way ANOVA $F_{(3,14)}$ = 136.7, p<0.0001; Dunnett's post hoc test control vs let-7c-5p or miR-128–3p: p=0.00000004 or 0.00000004, respectively). (**B**) The luciferase activity of the F2 reporter construct was unaffected by miR-125–5p (target position: 3248–3254) (one-way ANOVA $F_{(3,8)}$ = 5.59, p=0.023; Dunnett's post hoc test control vs let-7c-5p, miR-128–3 p or miR-125–5 p: p=0.5, p=0.07, or p=0.9, respectively). (**C**) Treatment with either miR-125–5 p (target position: 79–86) or let-7c-5p (target position: 166–173 and 262–269) decreased the luciferase activity of the positive control TRIM71 reporter construct (one-way ANOVA $F_{(3,8)}$ = 217.799, p<0.0001; Dunnett's post hoc test control vs let-7c-5p or miR-125–5 p: p=0.0000002 or p=0.0000002, respectively). (**D**) Left, luciferase activity was enhanced when cells were co-transfected with Let-7c-5p (2 nM) and its inhibitor (LNA-anti-let7c-5p, 25 pmol) compared to Let-7c-5p treatment alone (t-test, p=0.00003). Right, luciferase activity was increased when cells were co-transfected with miR128-3p (2 nM) and its inhibitor (LNA-anti- miR128-3p,25 pmol) compared to miR128-3p treatment alone (t-test, p=0.000006). (**E**) Top, schematics of WT and mutant sequences. Bottom, the effect of Let-7c-5p or miR128-3p treatment on reducing the luciferase activity of the WT F1 construct was prevented in the mutant F1 construct (let7c-5p: WT vs. Mut, t-test, p=0.00000005; miR 128–3p: WT vs. Mut, t-test, p=0.0001). *** indicates p<0.0005. Data are represented as mean ±SEM.

DOI: https://doi.org/10.7554/eLife.27872.009

object was significantly reduced in translin KO mice injected with eGFP virus (*Figure 7F2*). The viral restoration of translin/trax levels in adult excitatory neurons in the hippocampus rescued the memory deficits in KO mice (*Figure 7F2*). The lack of an effect of eGFP expression on the behavior of WT mice indicates that viral infection was not responsible for the observed rescue (*Figure 7F2*). The total object exploration time during training and testing did not differ across groups (*Figure 7—figure supplement 1A*, and *Figure 7—figure supplement 1B*; WT: 2.9 ± 0.3 s, KO$_{translin}$: 3.1 ± 0.6 s, KO$_{eGFP}$: 2.7 ± 0.2 s, WT$_{eGFP}$: 3.1 ± 0.4 s). In contrast to previous findings that male KO mice show

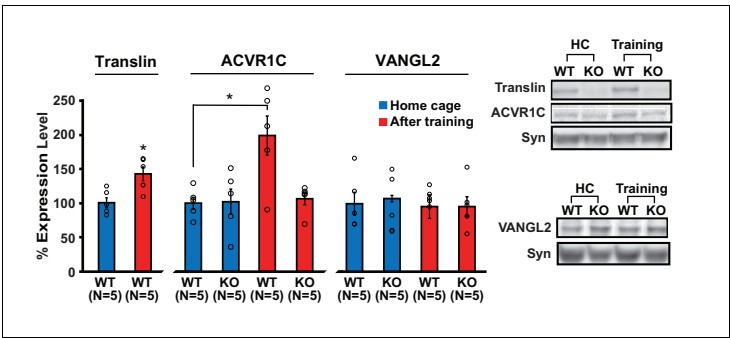

**Figure 5.** Loss of translin prevents elevation of ACVR1C protein levels 30 min following training in the object location memory task. Training induced an increase in translin levels in hippocampal synaptosomes (t-test, p=0.01), while the loss of translin prevented a training-induced upregulation of synaptosomal ACVR1C levels in the hippocampus (two-way ANOVA: genotype, $F_{(1,16)}$ = 5.2, p=0.03; training, $F_{(1,16)}$ = 6.8, p=0.02; genotype X training, $F_{(1,16)}$ = 5.7, p=0.03; Dunnett's post hoc test, $WT_{homecage}$ vs. $WT_{training}$: p=0.01, $WT_{homecage}$ vs. $KO_{training}$: p=0.9). However, synaptosomal vang-like 2 (VANGL2) levels were unaffected by either genotype or training (two-way ANOVA: genotype, $F_{(1,16)}$ = 0.06, p=0.8; training, $F_{(1,16)}$ = 0.3, p=0.6; genotype X training, $F_{(1,16)}$ = 0.06, p=0.8). Syn: synaptophysin. HC: homecage. * indicates p<0.05. n, number of mice. Data are represented as mean ±SEM.
DOI: https://doi.org/10.7554/eLife.27872.010

increased escape latency during trainings for the water maze task while female KO mice display enhanced freezing during fear memory testing (*Stein et al., 2006*), no difference was observed between male and female during object location memory task (*Figure 7—figure supplement 2*). Therefore, the function of translin/trax in adult excitatory neurons is critical for heterosynaptic strengthening of short-lived potentiation and long-term memory formation. Moreover, this viral reinstatement approach confirms that the observed deficits were not due to compensatory developmental changes caused by constitutive deletion of the translin gene.

## Discussion

Here, we investigated how translin/trax mediates learning-induced de novo synthesis of plasticity proteins that are critical for persistent synaptic plasticity and memory. Unlike conventional studies focusing on basal genetic alterations leading to defective phenotypes, we found that mice lacking translin/trax display increased levels of several microRNAs following learning, not at baseline. These microRNAs suppress the expression of the activin receptor ACVR1C, and the lack of translin/trax abolishes learning-induced synaptic upregulation of ACVR1C. Finally, inhibition of ACVR1C after synaptic stimulation or learning phenocopies the defects in synaptic plasticity or memory displayed by translin KO mice. Taken together, these findings provide compelling evidence for a novel pathway underlying long-term plasticity and memory, in which learning or synaptic stimulation engages translin/trax to reverse silencing of ACVR1C.

We found that translin KO mice display selective defects in synaptic plasticity and memory. Absence of translin/trax does not impact high frequency stimulation-induced long-lasting potentiation, but does cause selective impairment in heterosynaptic tagging. Behaviorally, although developmental and sex-specific alterations need to be considered, translin KO mice do not exhibit deficits in long-term spatial and contextual memory assessed by the water maze and fear conditioning (*Stein et al., 2006*). In contrast, we found in this study that both male and female translin KO mice exhibit long-term memory deficits in an object-location task. Furthermore, inhibition of ACVR1C, a target of translin/trax identified in the present study, selectively blocks the maintenance of synaptic tagging and object-location memory. Thus, object-location memory, in which mice integrate information about object identity and spatial location, may depend on the plasticity processes that mediate heterosynaptic plasticity.

This study is the first to show that ACVR1C is a key plasticity protein that is critical for heterosynaptic strengthening of short-lived plasticity and long-term memory, a finding that fits well with previous studies demonstrating that activin, a ligand for this receptor, plays a critical role in synaptic

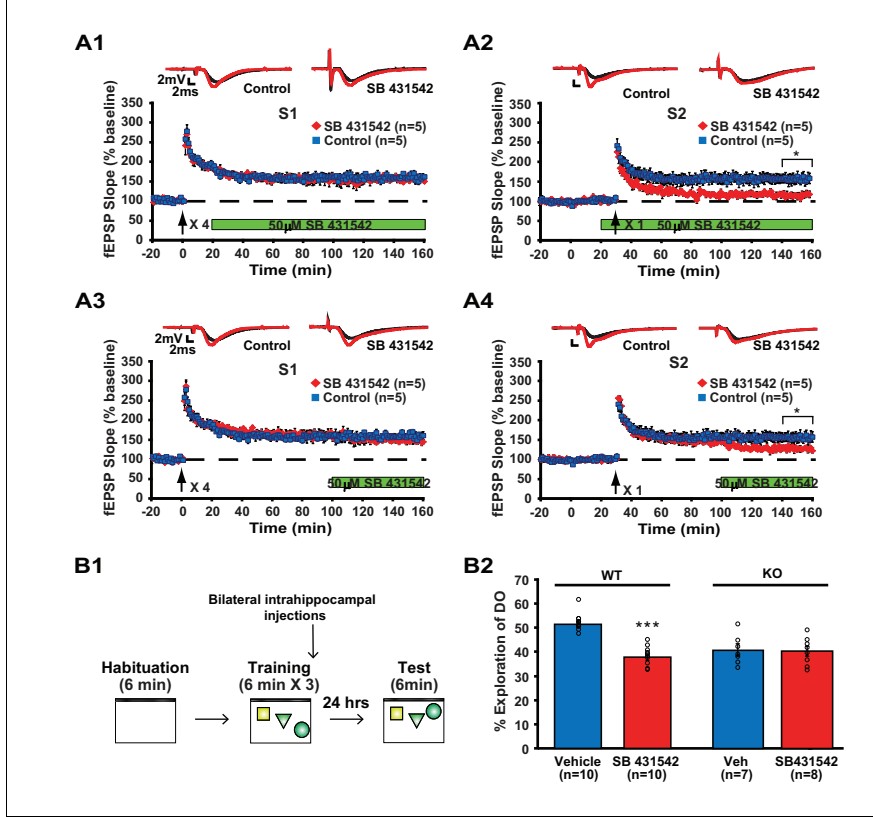

**Figure 6.** Inhibition of ACVR1C impairs synaptic tagging and long-term memory in WT mice, not translin KO mice. (**A**) In hippocampal slices from WT mice, treatment with SB431542 (50 μM), an inhibitor of ACVR1C, starting 10 min before S2 stimulation blocked persistent potentiation in pathway S2 (**A2**; two-way repeated-measures ANOVA, $F_{(1,8)} = 13.9$, p=0.007), without affecting 4-train long-lasting potentiation in pathway S1 (**A1**; two-way repeated-measures ANOVA, $F_{(1,8)} = 0.3$, p=0.6). Treatment with SB431542 during the last 1 hr of recording impaired the maintenance of S2 potentiation (**A4**; two-way repeated-measures ANOVA, $F_{(1,8)} = 6.1$, p=0.039), but not 4-train long-lasting potentiation in pathway S1 (**A3**; two-way repeated-measures ANOVA, $F_{(1,8)} = 1.4$, p=0.3). (**B1**) SB431542 or vehicle was bilaterally injected into the hippocampus immediately after training in the object-location memory task. (**B2**) While WT mice injected with SB431542 (1 μM) explored the DO significantly less than the vehicle-treated group, translin KO mice injected with SB431542 explored the DO at a level similar to the vehicle-treated group 24 hr after training (two-way ANOVA: genotype, $F_{(1,31)} = 9$, p=0.005; treatment, $F_{(1,31)} = 18.9$, p=0.0001; genotype X treatment, $F_{(1,31)} = 16.4$, p=0.0003; Tukey's post-hoc, WT$_{vehicle}$ vs. WT$_{SB431542}$: p<0.0001, KO$_{vehicle}$ vs. KO$_{SB431542}$: p=0.9). n, number of mice. * indicates p<0.05, *** indicates p<0.00005. Data are represented as mean ±SEM.

DOI: https://doi.org/10.7554/eLife.27872.011

The following figure supplement is available for figure 6:

**Figure supplement 1.** Related to *Figure 6*.
DOI: https://doi.org/10.7554/eLife.27872.012

plasticity and memory. Activin mRNA is upregulated by stimuli that induce long-term potentiation (*Inokuchi et al., 1996*), and activin treatment facilitates short-lived potentiation (*Ageta et al., 2010*). Furthermore, activin signaling is required for long-term memory formation (*Ageta et al., 2010*). Interestingly, we find that ACVR1C blockade selectively affects the maintenance of strengthened short-lived plasticity, a finding that distinguishes ACVR1C from other plasticity proteins postulated to be involved in synaptic tagging such as PKMζ and BDNF (*Barco et al., 2005*; *Sajikumar et al., 2009*; *Yao et al., 2008*). Although recent studies challenge the specificity of ZIP peptide in blocking PKMζ (*Volk et al., 2013*), treatment with ZIP impairs maintenance of both long-lasting plasticity and strengthened short-lived plasticity (*Sajikumar et al., 2009*; *Tsokas et al., 2016*; *Yao et al., 2008*). Additionally, deletion of BDNF blocks both induction and maintenance of strengthened short-lived

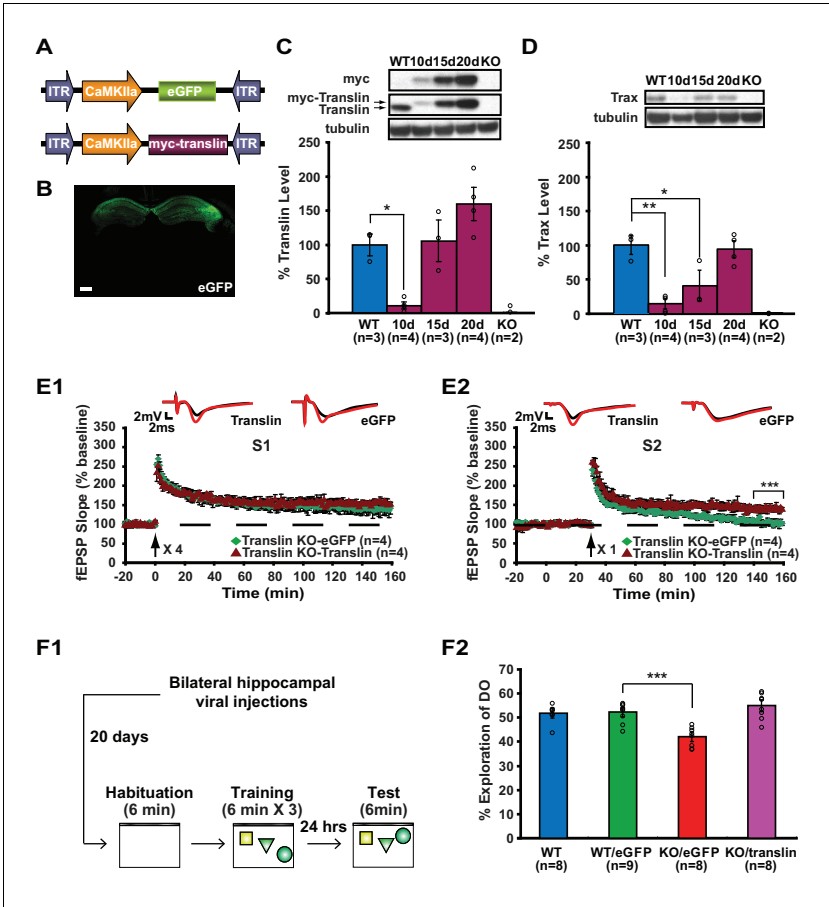

**Figure 7.** Viral restoration of translin/trax in hippocampal excitatory neurons rescues deficits in synaptic tagging and long-term memory in translin KO mice. (A) Viral constructs designs. The expression of myc-tagged translin and eGFP is driven by the CamKII promotor. (B) A representative image of hippocampal eGFP expression from a coronal brain section of adult translin KO mice 20 days after viral injection. (C) The level of virally expressed myc-tagged translin reached WT levels 15 to 20 days after viral injection into the hippocampus of adult translin KO mice (one-way ANOVA, $F_{(4,11)}$ = 15.4, p=0.0002, Dunnett's post hoc test, WT vs. $KO_{15day}$: p=0.9, WT vs. $KO_{20day}$: p=0.2). (D) Virally expressed translin restored trax to WT levels 20 days after injection (one-way ANOVA, $F_{(4,11)}$ = 14.7, p=0.0002, Dunnett's post hoc test, WT vs. $KO_{20day}$: p=0.9). (E) The viral restoration of translin levels in excitatory neurons of translin KO hippocampus reversed impaired persistent potentiation in pathway S2 (E2; two-way repeated-measures ANOVA, $F_{(1,6)}$ = 29.3, p=0.002) without affecting 4-train long-lasting potentiation in pathway S1 (E1; two-way repeated-measures ANOVA, $F_{(1,6)}$ = 2.5, p=0.2). (F1) A schematic diagram of the object-location memory task performed following bilateral hippocampal viral injections. (F2) Adult translin KO mice expressing eGFP explored the DO significantly less than WT littermates 24 hr after training. However, the behavior of adult translin KO mice expressing translin in excitatory neurons was similar to that of WT littermates or WT littermates expressing eGFP. Viral eGFP expression did not alter the behavior of WT littermates (one-way ANOVA, $F_{(3,29)}$ = 13.9, p=0.00001, Dunnett's post hoc test, $WT_{eGFP}$ vs. $KO_{eGFP}$: p=0.0001, $WT_{eGFP}$ vs. $KO_{translin}$: p=0.5, $WT_{eGFP}$ vs. WT: p=0.9). n, number of mice. Scale bar: 20 μm. * indicates p<0.05, ** indicates p<0.005, *** indicates p<0.0005. Data are represented as mean ±SEM.
DOI: https://doi.org/10.7554/eLife.27872.013

The following figure supplements are available for figure 7:

**Figure supplement 1.** Viral expression does not alter exploratory behavior during object-location memory task in translin KO mice.
DOI: https://doi.org/10.7554/eLife.27872.014

**Figure supplement 2.** Male and female mice show similar exploratory behavior during object-location memory task.
DOI: https://doi.org/10.7554/eLife.27872.015

plasticity (*Barco et al., 2005*). Overall, our results reveal the translin/trax/ACVR1C pathway as a new critical component of the molecular machinery mediating the maintenance of heterosynaptic plasticity and memory.

In current models of synaptic tagging, tags mark activated synapses and recruit locally translated proteins induced by strong synaptic input, which produces persistent potentiation of short-lived plasticity (*Barco et al., 2008*; *Park and Abel, 2015*; *Redondo and Morris, 2011*). Because phosphorylation of translin changes its RNA-binding affinity (*Kwon and Hecht, 1993*) and translin binds actin (*Wu et al., 1999*), it is plausible that synaptic tags could regulate the activity or localization of translin/trax, leading to reversal of translational silencing of ACVR1C at synapses receiving strong input. Indeed, we find that learning increases translin levels in synaptosomes supporting the view that it is recruited to the synapse after learning. Newly expressed ACVR1C could then be captured by tags generated at weakly activated synapses. At present, it is unclear how ACVR1C activation elicits persistent potentiation of weakly activated inputs. However, previous in vitro studies have demonstrated that activin is able to enhance synaptic contacts and spine growth via a process that requires CaMKII, PKA, ERK, and actin dynamics, all of which have been implicated as synaptic tags (*Barco et al., 2008*; *Hasegawa et al., 2014*; *Park and Abel, 2015*; *Redondo and Morris, 2011*; *Shoji-Kasai et al., 2007*).

We found that translin/trax suppresses certain microRNAs to elicit expression of ACVR1C only after training in a memory task, not at baseline. In other words, baseline levels of these miRNAs or ACVR1C and basal synaptic properties or behavior were unaffected in translin KO mice. Therefore, to avoid affecting baseline molecular pathways and behavior, we employed SB431542, a pharmacological inhibitor of ACVR1C shortly after training or synaptic stimulation. This temporal precision enabled us to demonstrate that ACVR1C inhibition can block maintenance even when applied 30 min following induction. In contrast, the slower time course of miRNA inhibitors or viral-based techniques capable of manipulating expression of ACVR1C would not be able to distinguish between effects on induction or maintenance. Moreover, constitutive inhibition of miR128, one of our target microRNAs, causes hyperactivity and premature death, and genetic mouse models targeting activin signaling showed altered glutamatergic transmission, locomotor activity and anxiety levels at baseline (*Ageta et al., 2008*; *Müller et al., 2006*; *Tan et al., 2013*). However, it is worth noting that SB431542 also inhibits ALK4 and ALK5 (*Inman et al., 2002*), receptors closely related to ACVR1C (also known as ALK7), and we cannot rule out the possibility that ALK4 and ALK5 play roles in memory formation. Future investigations including testing of inhibitors that specifically target ACVR1C when they become available would be important to confirm the role of ACVR1C in synaptic tagging and memory.

In summary, we have demonstrated that synaptic suppression of microRNA-mediated translational silencing is a novel mechanism mediating synaptic plasticity and memory. In doing so, we have identified translin/trax as a key enzyme activating this mechanism after synaptic stimulation and ACVR1C as a key plasticity product induced by reversal of silencing. BDNF also stimulates rapid translation of plasticity transcripts in cultured hippocampal neurons in vitro by suppressing silencing via activation of Lin28a, which targets microRNAs for degradation (*Huang et al., 2012*). Thus, activation of microRNA degrading enzymes may provide a general mechanism for driving translation in response to synaptic stimulation. As defects in the ability of synaptic stimulation to drive protein translation are thought to play a prominent role in neurodevelopmental and psychiatric disorders, we anticipate that elucidating this mode of translation regulation will yield valuable insights into the pathophysiology of these disorders.

## Materials and methods

### Translin knockout (KO) mice

Translin KO mice (MGI:2677496) were backcrossed to C57BL/6J (The Jackson Laboratory) for more than 15 generations. Heterozygous male and heterozygous female mice were mated to produce homozygous translin KO mice and WT littermates. Tail DNA was used for genotyping with 2 primers (GGC ATG GCA CAA ATA CTC AAG G and GTA GCC TTG TTG GAG TAC GTG) around the gene trap insertion site in intron 4, in which beta-geo is inserted, and 2 primers (CAA CGC TAT GTC CTG ATA GCG GTC C and CGT GTT CCG GCT GTC AGC GCA GG) around the inserted sequence. The

PCR products were one 500 bp band for homozygous KO mice and one 400 bp band for WT mice. Mice were maintained on a 12 hr light/12 hr dark cycle with lights on at 7 am (ZT0), and experiments were performed during the light cycle. Food and water were available *ad libitum*. Mice were randomly assigned to each experimental group. The number of mice used for behavioral and electrophysiological experiments was determined based on what has been used in the literature including those cited in this manuscript. All animal experiments were carried out in accordance with the NIH Guidelines and with the approval of the IACUC of the University of Pennsylvania, reference assurance number D16-00045 (A3079-01).

## Drug

SB431542 (Tocris, Cat. #: 1614), an inhibitor of activin receptor type 1C (*Inman et al., 2002*), was prepared as a 10 mM solution in 100% ethanol and delivered at 50 μM final concentration in artificial cerebrospinal fluid (aCSF, pH 7.4) for electrophysiology or at 1 μM final concentration in saline for cannulation experiments. Vehicle contained the same amount of 100% ethanol that was used to prepare the final concentration of SB431542.

## Antibody production

Translin and trax antibodies were produced (New England Peptide, Inc.) according to the sequences provided previously (*Finkenstadt et al., 2000*). The antibody synthesis was based on the C-terminal sequences of the human translin and trax, and the specificity was confirmed by Western blot. The peptide sequence for translin is CKYDLSIRGFNKETA, and for trax is CDVFSVKTEMIDQEEGIS.

## DNA manipulation and adeno-associated virus (AAV) constructs

The pAAV$_9$-CaMKIIα0.4-myc-translin and pAAV$_9$-CaMKIIα0.4-eGFP were produced through standard methods and packaged by the University of Pennsylvania viral core. Titers ranged from $1.06 \times 10^{13}$ to $2.02 \times 10^{13}$ genome copy numbers. The 0.4 kb CaMKIIα promoter fragment was used to drive expression selectively in excitatory neurons (*Dittgen et al., 2004*).

## Surgeries

Mice were anaesthetized using isoflurane and kept warm on a heating pad during the surgery. Buprenorphine and meloxicam were delivered to mice as analgesics (*Havekes et al., 2014*). Cannulation surgeries were done with 2- to 3-month-old male C57BL/6J mice or translin KO mice. Bilateral 22-gauge guide cannulae were held by dental glue (ESPE Ketac-Fil Plus Aplicap Glass Ionomer, 3M). The 33 gauge internal cannulae for the injections have a 0.5 mm projection. The coordinates were: anteroposterior, −1.9 mm, mediolateral,±1.5 mm, and 1.5 mm below bregma. Behavioral experiments started 1 week after surgery and an injection cannula was placed a day before the injection into the dorsal hippocampus. SB431542 (1 μM) or saline (vehicle) was loaded into a 5 μL Hamilton syringe and 1 μl was delivered to hippocampi bilaterally at 0.5 μl/min using a Harvard Apparatus Pump II Dual Syringe micropump. Injection cannulae remained in place for 1 min to allow the injected fluid to diffuse.

Viral injection was performed with male and female mice (2- to 3-month-old) using a nanofil 33G beveled needles (WPI) attached to a 10 μl Hamilton syringe controlled by a microsyringe pump (UMP3; WPI). The coordinates were: anteroposterior, −1.9 mm, mediolateral,±1.5 mm, and 1.5 mm below bregma. The needle was slowly lowered to the target site over the course of 3 min and remained at the target site for 1 min before beginning of the injection (0.2 μl per minute). Approximately 1 μl (corrected for genome copy number between constructs) was injected per hippocampus. After the injection, the needle remained at the target site for 1 min and then was slowly removed over a 5 min period. Experiments were performed 20 days after viral injection.

## Electrophysiology

Both male and female 2- to 4-month-old mice were sacrificed by cervical dislocation, and hippocampi were quickly collected in chilled, oxygenated aCSF (124 mM NaCl, 4.4 mM KCl, 1.3 mM MgSO$_4$·7H$_2$O, 1 mM NaH$_2$PO$_4$·H$_2$O, 26.2 mM NaHCO$_3$, 2.5 mM CaCl$_2$·2H$_2$O and 10 mM D-glucose) bubbled with 95% O$_2$/5% CO$_2$. A tissue chopper (Stoelting, Cat. #: 51425) was used to prepare transverse hippocampal slices (400 μm thick). Slices were placed in an interface recording chamber

at 28°C (Fine Science Tools, Foster City, CA), and aCSF was constantly perfused over slices at 1 ml/min. Slices were equilibrated for at least 2 hr in aCSF before beginning experiments. To record field excitatory postsynaptic potential (fEPSP), bipolar nichrome wire (0.5 mm, AM Systems, Carlsborg, WA) stimulating electrodes and a glass micropipette (1.5 mm OD, AM Systems, Carlsborg, WA) recording electrode filled with aCSF (resistance of 1–5 MΩ) were used. Slices having maximum fEPSP amplitude of less than 5 mV were rejected, and the stimulus strength was set to elicit 40% of the maximum fEPSP amplitude. Baseline values over the first 20 min were averaged, and this average was used to normalize each initial fEPSP slope. The input–output relationship was examined by measuring initial fEPSP slopes in response to increasing simulation intensity from 0 to 20 V with a 5 V increment. Paired-pulse facilitation (PPF) was measured by paired stimuli spaced 300, 200, 100, 50, and 25 ms apart.

For two-pathway experiments, massed 4-train (four 1 s 100 Hz trains delivered 5 s apart) stimulation that elicits long-lasting LTP was delivered to one pathway (S1) following 20 min baseline recordings. Thirty minutes later, 1-train (one 1 s 100 Hz train) stimulation that induces short-lived LTP was followed to another independent pathway (S2). PPF at 50 ms interval was used to confirm the independence of the two inputs.

## Object-location memory task

All animals (2- to 3 month old) were single housed one week before behavioral experiments. Sexes were balanced across groups, and littermates were used. Mice were handled for 3 min per day for six consecutive days prior to the experiment. All experiments were conducted between ZT0 and ZT2 as described previously (*Oliveira et al., 2010*). Briefly, mice underwent 6 min habituation session in an empty box followed by three 6 min training sessions in the same box containing three different objects (3.7 cm X 13 cm, 5.2 cm X 17 cm, 4 cm X 13 cm). The inter-session interval was 3 min, and mice were returned to their home cage after each session. An internal cue (three vertical black lines printed on a white paper (18 cm X 12 cm)) was attached on one wall of the box so that mice can locate each object relative to the cue when they were freely exploring the environment. After either 1 hr or 24 hr, mice were placed back in the training box in which the location of one of the objects was displaced. Time spent on the exploration of the displaced and non-displaced objects was hand-scored during the 6 min exploration period. The identity and location of each object were balanced between subjects. Exploration on an object was determined by sniffing, touching, and facing in a close proximity (within 1 cm). The experimenter was blind to the groups. For protein and microRNA assays, hippocampal tissue was collected 30 min after the last training session to probe for molecular mechanisms induced by learning to form long-term memory.

## Isolation of synaptosomes

Synaptosomes from hippocampal tissue were prepared as previously described (*Villasana et al., 2006*) with minor modifications. Briefly, hippocampi were collected from male translin KO mice and WT littermates (2- to 3-month-old) and homogenized using a TissueRuptor (Qiagen) in 1 ml lysis buffer (10 mM HEPES, 1 mM EDTA, 2 mM EGTA, 0.5 mM DTT, protease and phosphatase inhibitors). Hippocampal homogenates were gently sonicated with 3 pulses using an output power of 1 in a 60Sonic dismembrator (Fisher Scientific, Pittsburgh, PA). The samples were filtered twice through three layers of a pre-wetted 100 µm pore nylon filter (Millipore, Cat. #: NY 1H02500). The resulting filtrates were further filtered once through a pre-wetted 5 µm pore hydrophilic filter (Millipore, Cat. #: SMWP02500). After centrifugation at 1000 g for 15 min, pellets were resuspended in LDS sample buffer (Invitrogen, Cat. #: NP0007), and 10 µg of samples were loaded for Western blot analysis.

## Western blot analysis

Hippocampal tissue homogenization, protein separation, and transfer to polyvinylidene difluoride membranes were performed as previously described (*Havekes et al., 2012*). Membranes were blocked in 5% BSA or 5% non-fat milk in TBST and incubated with primary antibodies (translin, 1:100,000 for total hippocampal lysates, 1:1000 for synaptosomes; trax, 1:1,000; myc, Cell Signaling, Cat. #: 2276S (RRID:AB_331783), 1:5,000; ALK7, Millipore, Cat. #: 09–158 (RRID:AB_1163378), 1:1,000; VANGL2, Santa Cruz, Cat. #: sc-46561 (RRID:AB_2213082), 1:200) overnight at 4°C. Membranes were washed and incubated with appropriate horseradish peroxidase-conjugated goat anti-

mouse, anti-rabbit or donkey anti-goat IgG (Santa Cruz, 1:1,000) for 1 hr in room temperature. Blots were exposed on a film by ECL (Pierce, Cat. #: 32106) and quantified using ImageJ. The density of signal was normalized to β-tubulin levels (Sigma, Cat. #: T8328 (RRID:AB_1844090), 1:50,000) for total hippocampal lysates, or to synaptophysin levels (Millipore, Cat. #: MAB368 (RRID:AB_94947), 1:2,000) for synaptosomes. The mean protein level of each group was normalized to the mean protein level of the control WT group (100%).

## Immunohistochemistry

Translin KO mice that received intrahippocampal injection of eGFP virus were anesthetized with isoflurane and transcardially perfused with ice-cold PBS, followed by 4% paraformaldehyde in PBS. Brains were fixed in 4% paraformaldehyde at 4°C overnight and cryoprotected in 30% phosphate buffered sucrose for 3 days at 4°C. Coronal brain sections (20 μm) were made, mounted with gelatin (0.7%), and dried for 24 hr. The mounted sections were coverslipped with PermaFluor (Thermoscientific, Cat.#: TA-030-FM) and dried for 24 hr. Imaging was conducted on a Leica confocal microscope (TCS SP8).

For trax staining, rat hippocampal cultures were prepared from E18 embryos as previously described (*Wu et al., 2011*). At 7DIV, cultures were fixed with 4% formaldehyde in PBS and then processed for immunostaining with a rabbit polyclonal antibody to trax (1:5000) provided by Y. Chern (*Sun et al., 2006*) and human serum 18033 (1:3000) against GW182 provided by M. Fitzler. Antibodies were visualized by incubation with a mixture of anti-human Alexa 488 (1:1000) and anti-rabbit Alexa555 (1:2000). The specificity of the trax staining in rat hippocampal cultures has been confirmed by demonstrating that it is reduced by transfection with siRNA oligos that knock down trax expression but not by a mutant siRNA oligo that does not knock down trax expression (*Wu et al., 2011*).

## cDNA synthesis and quantitative real-time reverse transcription (RT)-PCR

Hippocampi from male translin KO mice and WT littermates (2- to 3-month-old) were collected 30 min after training in the object-location memory task, homogenized in 1 ml of Trizol (Invitrogen, Carlsbad, CA), and incubated at room temperature for 5 min. Samples were transferred to phase-lock gel tubes (Eppendorf, Westbury, NY) containing chloroform (140 μl) followed by vigorous mixing and room temperature incubation for 3 min. After centrifugation at 4°C at full speed for 15 min, the aqueous phase was transferred to new tubes, in which 525 μl of 100% ethanol was added. RNA was purified using the RNeasy system (Qiagen, Valencia, CA) according to the protocol of the manufacturer. Residual DNA was removed by treatment with DNA-free (Ambion). For microRNA assays, 250 ng RNA was used in each miScript microRNA PCR Arrays (Qiagen). cDNA synthesis with miScript HiSpec Buffer was performed according to the manufacturer's protocol. cDNA reactions were diluted in 200 μl of RNase-free water. A neurological development and disease microRNA PCR array (MIMM-107ZE-1, Qiagen) was used to probe for pathophysiologically relevant candidate microRNAs affected in translin KO mice that underwent the object-location memory task. The experiment was performed in duplicate according to the manufacturer's protocol with cDNA samples from a pair of translin KO mouse and WT littermate. This probe test gave 9 candidate microRNAs that could be affected in translin KO mice after training, and these microRNAs were verified with real-time RT-PCR reactions that were prepared in 384-well optical reaction plates with optical adhesive covers (ABI, Foster City, CA). Each reaction was composed of 1 μl cDNA, 5 μl 2x Quantitect SYBERGreen Master Mix (Qiagen), 1 μl Universal primer, 2 μl RNase-free water, and 1 μl of one of the following miScript Primers (Qiagen): let 7b-5p – MS00001225, let 7c-5p – MS00005852, let 7d-5p – MS00001232, let 7e-5p – MS00032186, miR 124–3p – MS00029211, miR 125b-5p – MS00005992, miR128-3p – MS00011116, miR 9–5p – MS00012873, miR 9–3p – MS00005887, miR 409-3p – MS00011970, SNORD68 – MS00033712. Reactions were performed in duplicate on the Viia7 Real-Time PCR system (Life Technologies, Carlsbad, CA). This real-time RT-PCR experiment was also performed with hippocampal cDNA samples collected from translin KO mice and WT littermates (2- to 3-month-old) that remained in the home cage (handling-only). The ΔΔCt method was used for relative quantification of gene expression. The mean expression level of translin KO groups was normalized to the mean expression level of the control WT group. For bioinformatics analyses, online database

TargetScan was used to search for the predicted targets (Refseq IDs) of the validated microRNAs that were affected in translin KO mice after training. Specifically, the list of target genes from human and mouse was obtained from TargetScan. We then compared the human and mouse list to find conserved targets of each microRNA between the two species. Finally, the comparison between the conserved targets of each microRNAs yielded their common targets, and context scores provided by TragetScan were used to estimate the strength of target prediction.

The algorithm used to find common targets of let7-c, miR128 and miR125b is provided below.

```perl
use strict;
my $targets1= @ARGV[0];
my $targets2= @ARGV[1];
my $outfile="Overlap_".$targets1.$targets2;
my %ref;
open IN, "<$targets1" or die $!;
while (<IN>){
    if (/(NM_\d+)/){
        $ref{$1}=$_;
    }
}
my $i=0;
open IN2, "<$targets2" or die $!;
open OUT, ">$outfile" or die $!;
while (<IN2>){
    if (/(NM_\d+)/){
        my $ID=$1;
        my $match=$ref{$ID};
        if (defined $match){
            $i++;
            print OUT "$ID\n";
        }
    }
}
print " NUmber of overlaps between $targets1 and $targets2 is $i\n";
```

## Luciferase reporter assays

*ACVR1C* 3'UTR fragments were PCR amplified from mouse genomic DNA using the following primers: ACVR1C-3'UTR-F1 sense 5'cagctgtgtgtcaaggaagactgt3'; antisense 5'agttgtcacagggttcgtaacc3'. ACVR1C-3'UTR-F2 sense 5'gtatgcatccttccacgtct3'; antisense 5'gatgactgtcttcactaagac3'. These fragments were ligated into pGEM-T easy (Promega) according to the manufacturer's guidelines, and then inserted into a single *NotI* site in frame with the 3' end of luciferase in the psiCHECK-2 reporter plasmid (Promega). Fragment orientation was verified by sequencing. The mouse TRIM71 3'UTR luciferase reporter was purchased from Genecopeia (Rockville, MD).

An aliquot of HEK 293 cells (RRID:CVCL_0045) were gifted from the Snyder lab at Johns Hopkins School of Medicine, which obtained the cells from ATCC. Although this specific batch was not tested for mycoplasma contamination, there was no evidence of contamination, such as reduced proliferation rate or morphological changes. HEK293 cells were used because they are well-known and widely used for their favorable transfection properties. HEK293 cells grown in 24-well-plates to 70–90% confluence were transfected using Lipofectamine 2000 (Life Sciences) with 150 ng of one of the psiCHECK-2 reporter plasmids along with 2 nM of one of the microRNA mimics (Dharmacon). Control wells received only reporter plasmid. Twenty-four hours later, cells were lysed and luciferase activities were monitored with the Dual-Luciferase assay according to the manufacturer's guidelines (Promega). For miRNA inhibitor experiments LNA-anti-miR128-3p (4101052–002) and LNA-anti-let7c-5p (4100669–002) were purchased from Exiqon.

## Data analysis

Data analyses were performed using Statistica 10 or SPSS V10. The LTP data were analyzed using a repeated-measures ANOVA test on the last 20 min of the initial fEPSP slope values normalized to the average baseline value. To analyze input–output data, a t-test was used to compare the average linear regression slopes between each group. PPF data were analyzed using a two-way repeated-measures ANOVA. For evaluation of biochemical, behavioral, and gene expression data, a t-test or a two-way ANOVA was performed. Dunnett's post hoc test was performed if applicable. Differences were considered statistically significant when $p < 0.05$. Data were plotted as mean ±S.E.M.

## Acknowledgements

This work was supported by the Kwanjeong Educational Foundation (to A.P.), Netherlands Organization for Scientific Research NWO-Rubicon Grant 825.07.029 (to R.H.), DA-000266 (to J.B.) and NIH Grant R01 MH087463 (to T.A.). TA is also supported by the Brush Family Chair of Biology at the University of Pennsylvania and the Roy J. Carver Chair of Neuroscience at the University of Iowa. The authors declare no conflicts of interest. We thank Giulia Porcari and Morgan Bridi for help with manuscript preparation.

## Additional information

### Funding

| Funder | Grant reference number | Author |
|---|---|---|
| The Kwanjeong Educational Foundation | Graduate Student Fellowship | Alan Jung Park |
| Nederlandse Organisatie voor Wetenschappelijk Onderzoek | Postdoctoral Fellowship | Robbert Havekes |
| National Institutes of Health | MH087463 | Ted Abel |
| National Institutes of Health | DA-000266 | Jay Baraban |
| Brush Family Chair of Biology at the University of Pennsylvania | Faculty Award | Ted Abel |
| Roy J. Carver Chair of Neuroscience at the University of Iowa | Faculty Award | Ted Abel |

The funders had no role in study design, data collection and interpretation, or the decision to submit the work for publication.

### Author contributions

Alan Jung Park, Conceptualization, Data curation, Formal analysis, Supervision, Funding acquisition, Validation, Investigation, Methodology, Writing—original draft, Project administration, Writing—review and editing; Robbert Havekes, Supervision, Methodology; Xiuping Fu, Jennifer C Tudor, Zhi Li, Yen-Ching Wu, Data curation, Formal analysis; Rolf Hansen, Data curation; Lucia Peixoto, Software, Formal analysis; Shane G Poplawski, Data curation, Methodology; Jay M Baraban, Conceptualization, Resources, Supervision, Validation, Writing—review and editing; Ted Abel, Conceptualization, Resources, Supervision, Funding acquisition, Validation, Writing—review and editing

### Author ORCIDs

Alan Jung Park, https://orcid.org/0000-0002-7644-8886
Robbert Havekes, https://orcid.org/0000-0003-0757-4739
Jennifer C Tudor, https://orcid.org/0000-0002-3826-3012
Jay M Baraban, https://orcid.org/0000-0002-8165-2638
Ted Abel, http://orcid.org/0000-0003-2423-4592

## Ethics

Animal experimentation: All animal experiments were carried out in accordance with the NIH Guidelines and with the approval of the IACUC of the University of Pennsylvania, reference assurance number D16-00045 (A3079-01).

## Decision letter and Author response

Decision letter https://doi.org/10.7554/eLife.27872.017
Author response https://doi.org/10.7554/eLife.27872.018

## Additional files

### Supplementary files

• Transparent reporting form
DOI: https://doi.org/10.7554/eLife.27872.016

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
