## [Decision Letter]

Thank you for submitting your article "Learning induces translin/trax RNase complex to express activin receptors for persistent memory" for consideration by *eLife*. Your article has been reviewed by three peer reviewers, one of whom, Lisa M Monteggia (Reviewer #1), is a member of our Board of Reviewing Editors, and the evaluation has been overseen by Eve Marder as the Senior Editor.

The reviewers have discussed the reviews with one another and the Reviewing Editor has drafted this decision to help you prepare a revised submission.

The reviewers are all agreed on the merits and significance of the work. However, they also concur that the manuscript requires major re-editing and clarification to avoid the confusion generated by the current organization of the study. Despite the extensive nature of the revision, the reviewers all agreed that the revised manuscript does not necessarily require incorporation of new experiments. Since there are no new experimental requirements please refer to the individual reviews to address the editorial revisions needed.

Reviewer #1

In the manuscript titled "Learning induces translin/trax RNAse complex to express activin receptors for persistent memory," the authors propose that following stimulation which induces heterosynaptic plasticity the translin/trax RNAse complex is trafficked to the post-synapse where it degrades certain miRNAs causing the disinhibition of translation of certain mRNAs. They suggest this disinhibition is necessary for the long term maintenance of heterosynaptic plasticity. Specifically the authors posit that the degradation of miR-128-3p and let-7c-5p to disinhibit the activin receptor ACVR1C is crucial for maintaining heterosynaptic plasticity and behavioral memory in the displaced object test. The data provided by the authors makes a strong case for their hypothesis but there are a few concerns that should be addressed.

Comments:

1) The authors should present data if they have it on whether the miRNAs which have altered profiles after training (let-7c-5p, miR-125b-5p, miR-128-3p, and miR-9-3p) can be degraded by translin/trax in vitro. This would significantly strengthen the study.

2) Figure 3 home cage and after training data should be plotted together and normalized to home cage WT (as in Figure 5) to highlight that the alteration in miRNA level is caused by a decrease in miRNAs in WT after training that is not seen in KO rather than an increase in KO miRNAs above WT levels after training.

3) Figure 6B2 behavioral data does not seem to replicate Figure 1 data. Specifically, the KO mice do not look like they have decreased exploration of the displaced object compared to WT mice. Please comment/clarify this point.

4) There is a question whether Figure 5 ACVR1C and VANGL2 should be analyzed by a repeated measures two-way ANOVA (genotype x before/after training) to show an interaction effect for ACVR1C. The same comment on whether Figure 6B2 (genotype x drug) and 7F2 (genotype x virus) should also be analyzed by two-way ANOVAs.

5) As a minor note, there appears to be a textual error in paragraph two of subsection “Translin/Trax RNAse complex reverses microRNA silencing of ACVR1C transcript after learning”. Figure 4 is a positive control not the ACVR1C fragment with the miR-125b-5p target site.

Reviewer #2:

This manuscript describes a logical series of experiments demonstrating a role for translin/trax in activity-dependent down-regulation of miRNA (particularly let-7c-5p and miR-128-3p) in a way that upregulates activin receptor 1C. Both translin and ACVR1C were shown to be required for normal synaptic tag and capture (STC)-based LTP and long-term memory for object location. Importantly, deficits in each caused by translin-KO could be rescued by AAV-mediated restoration of translin levels. The demonstration of an activity-dependent reversal of miRNA silencing that is important for plasticity and behaviour is an important step forward for this field.

The authors have done an excellent job of pulling out a phenotype of the translin-KO mice, as they have previously reported limited effects on water maze learning and no impairment in fear conditioning. The effect here is thus subtle. The mice appear to explore the objects only minimally on the test day (<4 s out of 6 min), and out of that time the displaced object is explored dropped from 52% to 42% in the KO mice. This subtlety was reflected in the electrophysiology where S1 LTP was not affected by the KO or the drug, but priming of S2 LTP was. It would be good to have more discussion as to why translin appears to play no role under stronger learning or LTP protocols. I wasn't particularly convinced, for example, that the object location task was any more analogous to an STC protocol than any of the other behavioural tasks, in that all involve some kind of associative interaction.

It is curious that translin KO had no effect on home-cage miRNA levels. Please comment on this. More importantly, the authors suggest that after learning, translin mediates a rapid degradation of miRNA. It would be helpful to provide evidence that this actually occurs. Are the levels of any of the tested miRNA reduced compared to home-cage controls in WT mice? Is translin phosphorylated after training in WT mice? Such data would help solidify the concept that translin is activated after training, adding to its increase in synaptosomes which is perhaps less diagnostic.

Reviewer #3:

In this very interesting study the authors provide outstanding new information on a complex consisting of two RNA binding proteins, translin/trax. Until recently, this protein complex has been implicated in various processes including synaptic plasticity, learning and memory, however, it was not considered as one of the major players in the field. Since a recent publication by one of the authors, Dr. Baraban, in Cell Reports in 2014, there was a change in perception as the protein complex had been implicated as suppressor of miRNA-mediated translational silencing. Consequently, the authors investigate a previously published translin ko mice (also lacking trax due to its own instability) for deficits in synaptic plasticity and long-term memory. Importantly, these ko mice display specific defects in long-term memory and synaptic tagging, a form of heterosynaptic plasticity implicated in memory storage. In a pilot experiment, the authors then identified a few miRNAs that are significantly increased in translin ko mice after training and searched for conserved targets that might be regulated by these three key miRNAs, i.e. let-7c, miR-125b-5p and miR-128-3p. From the short listed targets, the authors identified one promising candidate, coding for activin receptor 1C (ACVR1C, or activin-like kinase 7, ALK7). Luciferase reporter assays then showed that ACVR1C mRNA is indeed regulated by two of these miRNAs. A general ALK inhibitor selectively blocked heterosynaptic strengthening of one pathway (S2) in hippocampal slices from wt mice, but not in translin ko mice. In line with these findings is the finding that the ALK inhibitor impaired long-term novel object location memory (a form of memory that is hippocampus dependent) 24 hours after training in wt but not translin ko mice. Finally, and most impressively, the authors were able to rescue the observed deficits by intrahippocampal re-expression of translin via AAV.

In my opinion, the authors not only substantiate the new molecular function of translin/trax as a miRNase, but provide stunning new data for the in vivo role of the complex in specific forms of synaptic plasticity and learning and memory. Before this important, original and timely study can be published, there is serious effort needed to bring this quite unusual manuscript in a form that suits the standard for a major journal. Given the fact that most authors have left their laboratories at the time of the study and the often-incoherent red line, I have the impression that this manuscript has been in the peer review process since a long time having suffered major (inconsistent) revisions. I would strongly encourage the authors to give proper background information on the existing data: the published translin ko mice, their correct molecular, behavioral and electrophysiological phenotype, cite key papers from the authors' labs (Stein 2006), explain the observed female versus male differences and how this is consistent with the new study, changes in the interpretation of the function of the translin/trax complex over the last 18 years or so, define better what the authors mean by the "translin/trax RNase complex" and what does this precisely mean molecularly. Accordingly, I would suggest to redo and extend the Introduction. Moreover, the Results section is inadequate in terms of guiding the reader through the experiments and key experiments have to be properly explained in the text (e.g. Figure 2, what is actually shown here? Include size markers. Label the visible bands and symbols; indicate Figure 3 as qPCR experiment in the text, Figure 3: how was the miRNA screening performed?), and figure legends are insufficient, sometimes misleading, and not properly written as they interpret the data instead of explain the techniques/methods.

There are key experimental issues that must be fixed before publication. First, two miRNAs (let-7, miRNA 128) are essential for this story (see above), but why have the authors chosen miR-409 for the miRNA cleavage assays in vitro, although it is not relevant at all in this context? Second, another crucial experiment is shown in synaptoneurosomes in Figure 5. Throughout the manuscript, the authors carefully distinguish between short-term and long-term plastic effects, but why did they decide to investigate translin protein levels after 30 min of learning (= short-term). Shouldn't be there only an effect after 24h, but NOT after only 30 min, as demonstrated in all the other experiments (no effect after 1 hour)? These fractionations should be performed after at least 1h post-training. Third, the authors must investigate and comment on whether the other members of the activin receptor (ALK4, ALK5) are possibly regulated by the same set of miRNAs than ALK7, as the inhibitor is less potent for ALK7 than for the other ALK members. Have the authors checked the expression levels of ALK4 and 5 in the hippocampus? Forth, in their rescue experiment presented in Figure 7 F2, are the levels for the three key miRNAs back to normal levels in trained animals? Finally, the Abstract is not doing justice to the outstanding data presented but contain more speculation than own data.

---

## [Author Response]

Reviewer #1In the manuscript titled "Learning induces translin/trax RNAse complex to express activin receptors for persistent memory," the authors propose that following stimulation which induces heterosynaptic plasticity the translin/trax RNAse complex is trafficked to the post-synapse where it degrades certain miRNAs causing the disinhibition of translation of certain mRNAs. They suggest this disinhibition is necessary for the long term maintenance of heterosynaptic plasticity. Specifically the authors posit that the degradation of miR-128-3p and let-7c-5p to disinhibit the activin receptor ACVR1C is crucial for maintaining heterosynaptic plasticity and behavioral memory in the displaced object test. The data provided by the authors makes a strong case for their hypothesis but there are a few concerns that should be addressed.Comments:1) The authors should present data if they have it on whether the miRNAs which have altered profiles after training (let-7c-5p, miR-125b-5p, miR-128-3p, and miR-9-3p) can be degraded by translin/trax in vitro. This would significantly strengthen the study.

A previous study (Asada et al., 2014) demonstrated that recombinant translin/trax is able to degrade pre-microRNAs in vitro, using pre-miR-16 and pre-miR-21 as prototypical substrates. Furthermore, this study showed that this activity depended on the presence of mismatches in the stem, as removing those abolished cleavages by translin/trax in vitro. Because all four of the microRNAs shown to be elevated following training in translin KO mice contain similar or more extensive mismatches in the stem, we feel it is reasonable to assume that pre-microRNAs of let-7c-5p, miR-125b-5p, miR-128-3p, and miR-9-3p can be cleaved by translin/trax in vitro. Accordingly, we have not conducted these studies and do not have in vitro data showing that translin/trax degrades let-7c-5p, miR-125b-5p, miR-128-3p, and miR-9-3p.

2) Figure 3 home cage and after training data should be plotted together and normalized to home cage WT (as in Figure 5) to highlight that the alteration in miRNA level is caused by a decrease in miRNAs in WT after training that is not seen in KO rather than an increase in KO miRNAs above WT levels after training.

We agree with the advantage of presenting Figure 3 according to the reviewer’s suggestion. Unfortunately, however, sample collection and qPCR assays performed for the home cage and after training conditions were done as separate experiments, which makes it undesirable to combine them. Specifically, qPCR master mix and qPCR plates were independently prepared for home cage and post-training conditions, and there is no appropriate way to correct for potential differences between the two assays. Hence, it would be inappropriate to normalize training data with home cage WT data. Rather, our data show that training increases target miRNA levels in KO compared to WT mice.

3) Figure 6B2 behavioral data does not seem to replicate Figure 1 data. Specifically, the KO mice do not look like they have decreased exploration of the displaced object compared to WT mice. Please comment/clarify this point.

Thank you for finding this error. There was a scaling mistake when we combined the KO bar graph with the WT bar graph. Originally, the WT and KO bar graphs were presented as separate figures. This scaling mistake is also apparent when comparing values in the text; “(Figure 6B2; vehicle: 40.6 ± 2.5%; SB 431542: 40.1 ± 2.1%)” with the height of the bar graph for KOs (around 46%). We apologize for this mistake, and we thank the reviewer for identifying this. Figure 6B2 is now fixed so that the proper bars, which match the values in the text, are now in Figure 6B2.

4) There is a question whether Figure 5 ACVR1C and VANGL2 should be analyzed by a repeated measures two-way ANOVA (genotype x before/after training) to show an interaction effect for ACVR1C.

We cannot perform a repeated-measures test because protein levels were measured only once, not repeatedly, before or after training, Instead, we ran a two-way ANOVA tests. For ACVR1C, two-way ANOVA shows the main effect of genotype (p=0.03) or training (0.02), and interaction (p=0.03). For VANGL2, there is no effect of genotype, training, or interaction (p>0.6). This information is provided in the figure legend.

The same comment on whether Figure 6B2 (genotype x drug) and 7F2 (genotype x virus) should also be analyzed by two-way ANOVAs.

For Figure 6B2, there is the main effect of genotype (p=0.005) or drug (p=0.0001) and a significant interaction (p=0.0003). We could not run a two-way ANOVAs for Figure 7F2 because not all groups had the same virus treatments. In other words, WT alone group did not get a viral injection and only two groups (WT/eGFP and KO/eGFP) received eGFP viral injection, while only KO/translin group had translin viral injection. We have added these statistics in the figure legend.

5) As a minor note, there appears to be a textual error in paragraph two of subsection “Translin/Trax RNAse complex reverses microRNA silencing of ACVR1C transcript after learning”. Figure 4 is a positive control not the ACVR1C fragment with the miR-125b-5p target site.

We corrected the text as following. “The luciferase activity of the positive control TRIM71, which has target sites for let-7c-5p and miR-125-5p, was suppressed by let-7c-5p and miR-125-5p, but not miR-128-3p (Figure 4)”.

Reviewer #2:This manuscript describes a logical series of experiments demonstrating a role for translin/trax in activity-dependent down-regulation of miRNA (particularly let-7c-5p and miR-128-3p) in a way that upregulates activin receptor 1C. Both translin and ACVR1C were shown to be required for normal synaptic tag and capture (STC)-based LTP and long-term memory for object location. Importantly, deficits in each caused by translin-KO could be rescued by AAV-mediated restoration of translin levels. The demonstration of an activity-dependent reversal of miRNA silencing that is important for plasticity and behaviour is an important step forward for this field.The authors have done an excellent job of pulling out a phenotype of the translin-KO mice, as they have previously reported limited effects on water maze learning and no impairment in fear conditioning. The effect here is thus subtle. The mice appear to explore the objects only minimally on the test day (<4 s out of 6 min), and out of that time the displaced object is explored dropped from 52% to 42% in the KO mice. This subtlety was reflected in the electrophysiology where S1 LTP was not affected by the KO or the drug, but priming of S2 LTP was. It would be good to have more discussion as to why translin appears to play no role under stronger learning or LTP protocols. I wasn't particularly convinced, for example, that the object location task was any more analogous to an STC protocol than any of the other behavioural tasks, in that all involve some kind of associative interaction.

We agree that it is better to provide more discussion on this matter, and we have added the following paragraph as the second paragraph of the Discussion.

“We found that translin KO mice display selective defects in synaptic plasticity and memory. Absence of translin/trax does not impact high frequency stimulation-induced long-lasting potentiation, but does cause selective impairment in heterosynaptic tagging. Behaviorally, although developmental and sex-specific alterations need to be considered, translin KO mice do not exhibit deficits in long-term spatial and contextual memory assessed by the water maze and fear conditioning (Stein et al., 2006). In contrast, we found in this study that both male and female translin KO mice exhibit long-term memory deficits in an object-location task. Furthermore, inhibition of ACVR1C, a target of translin/trax identified in the present study, selectively blocks the maintenance of synaptic tagging and object-location memory. Thus, object-location memory, in which mice integrate information about object identity and spatial location, may depend on the plasticity processes that mediate heterosynaptic plasticity”.

It is curious that translin KO had no effect on home-cage miRNA levels. Please comment on this.

This is one of the most interesting findings of the present study. Our working model posits that training activates translin/trax to degrade selected miRNAs, so translin KO mice display elevations in these miRNAs relative to WT mice following training. However, under baseline conditions, translin/trax is inactive, so levels of these miRNAs are unchanged between WT and KO mice. We have emphasized this point at appropriate places in the manuscript. We also added a statement in the first paragraph of the Discussion: “Unlike conventional studies focusing on basal genetic alterations leading to defective phenotypes, we found that mice lacking translin/trax display increased levels of several microRNAs following learning, not at baseline”.

More importantly, the authors suggest that after learning, translin mediates a rapid degradation of miRNA. It would be helpful to provide evidence that this actually occurs. Are the levels of any of the tested miRNA reduced compared to home-cage controls in WT mice?

Please refer to our response to the reviewer 1 comment 2.

Is translin phosphorylated after training in WT mice? Such data would help solidify the concept that translin is activated after training, adding to its increase in synaptosomes which is perhaps less diagnostic.

At present, the mechanism mediating the activation of translin/trax RNase activity is not known. Certainly, phosphorylation is a leading possibility and will be the subject of future investigations once phospho-translin antibodies become available.

Reviewer #3:In this very interesting study the authors provide outstanding new information on a complex consisting of two RNA binding proteins, translin/trax. Until recently, this protein complex has been implicated in various processes including synaptic plasticity, learning and memory, however, it was not considered as one of the major players in the field. Since a recent publication by one of the authors, Dr. Baraban, in Cell Reports in 2014, there was a change in perception as the protein complex had been implicated as suppressor of miRNA-mediated translational silencing. Consequently, the authors investigate a previously published translin ko mice (also lacking trax due to its own instability) for deficits in synaptic plasticity and long-term memory. Importantly, these ko mice display specific defects in long-term memory and synaptic tagging, a form of heterosynaptic plasticity implicated in memory storage. In a pilot experiment, the authors then identified a few miRNAs that are significantly increased in translin ko mice after training and searched for conserved targets that might be regulated by these three key miRNAs, i.e. let-7c, miR-125b-5p and miR-128-3p. From the short listed targets, the authors identified one promising candidate, coding for activin receptor 1C (ACVR1C, or activin-like kinase 7, ALK7). Luciferase reporter assays then showed that ACVR1C mRNA is indeed regulated by two of these miRNAs. A general ALK inhibitor selectively blocked heterosynaptic strengthening of one pathway (S2) in hippocampal slices from wt mice, but not in translin ko mice. In line with these findings is the finding that the ALK inhibitor impaired long-term novel object location memory (a form of memory that is hippocampus dependent) 24 hours after training in wt but not translin ko mice. Finally, and most impressively, the authors were able to rescue the observed deficits by intrahippocampal re-expression of translin via AAV.In my opinion, the authors not only substantiate the new molecular function of translin/trax as a miRNase, but provide stunning new data for the in vivo role of the complex in specific forms of synaptic plasticity and learning and memory. Before this important, original and timely study can be published, there is serious effort needed to bring this quite unusual manuscript in a form that suits the standard for a major journal. Given the fact that most authors have left their laboratories at the time of the study and the often-incoherent red line, I have the impression that this manuscript has been in the peer review process since a long time having suffered major (inconsistent) revisions. I would strongly encourage the authors to give proper background information on the existing data: the published translin ko mice, their correct molecular, behavioral and electrophysiological phenotype, cite key papers from the authors' labs (Stein 2006), explain the observed female versus male differences and how this is consistent with the new study, changes in the interpretation of the function of the translin/trax complex over the last 18 years or so, define better what the authors mean by the "translin/trax RNase complex" and what does this precisely mean molecularly. Accordingly, I would suggest to redo and extend the Introduction. Moreover, the Results section is inadequate in terms of guiding the reader through the experiments and key experiments have to be properly explained in the text (e.g. Figure 2, what is actually shown here? Include size markers. Label the visible bands and symbols; indicate Figure 3 as qPCR experiment in the text, Figure 3: how was the miRNA screening performed?), and figure legends are insufficient, sometimes misleading, and not properly written as they interpret the data instead of explain the techniques/methods.

We have revised the manuscript as suggested. In particular, we have expanded the Introduction to include additional background on the phenotypes observed previously in these mice, the advances our study made in our understanding the function of the translin/trax, and a detailed description of the molecular composition of the translin/trax RNase complex. Additional details about these changes are detailed below. The male vs. female differences were described in paragraph two of subsection “Restoration of translin/trax to physiological levels in hippocampal excitatory neurons rescues deficits in synaptic tagging and long-term memory”.

Regarding Figure 3, we indicated qPCR in the text (paragraph two subsection “Loss of translin/trax alters microRNA levels after learning”) and the legend. For Figure 3, more detailed description about the miRNA screening was provided in the Results and in the Materials and methods under “cDNA synthesis and quantitative real-time reverse transcription (RT)-PCR”.

There are key experimental issues that must be fixed before publication. First, two miRNAs (let-7, miRNA 128) are essential for this story (see above), but why have the authors chosen miR-409 for the miRNA cleavage assays in vitro, although it is not relevant at all in this context?

We chose miR-409 to illustrate the RNase activity of translin/trax because both our group and another group (Asada et al., 2014) independently found that the level of miR-409 is elevated in the brain of translin KO mice. As we agree that miR-409 cleavage data are irrelevant to our study, we omitted these data (and Figure 2) from the revised manuscript.

Second, another crucial experiment is shown in synaptoneurosomes in Figure 5. Throughout the manuscript, the authors carefully distinguish between short-term and long-term plastic effects, but why did they decide to investigate translin protein levels after 30 min of learning (= short-term). Shouldn't be there only an effect after 24h, but NOT after only 30 min, as demonstrated in all the other experiments (no effect after 1 hour)? These fractionations should be performed after at least 1h post-training.

Our study indicates that learning induces translin-mediated synaptic expression of activin receptors to produce long-lasting synaptic plasticity and long-term memory. In other words, we focused on a mechanism activated immediately after learning to produce long-term memory, a mechanism independent of that for short-term memory. Indeed, treatment with the activin inhibitor SB431542 immediately after learning impaired long-term memory (Figure 6), and translin KO mice display impairments for long-term, not short-term memory (Figure 1). Moreover, delivering SB431542 during or 1-hour after the tagging stimulation impaired synaptic tagging (Figure 6). These findings suggest that the activin signaling activated within 1-hour after learning is critical for long-term memory formation. Hence, we collected hippocampal tissue samples 30 minutes after learning. We have added “For protein and microRNA assays, hippocampal tissue was collected 30 minutes after the last training session to probe for molecular mechanisms induced by learning to form long-term memory.” in the Materials and methods.

Third, the authors must investigate and comment on whether the other members of the activin receptor (ALK4, ALK5) are possibly regulated by the same set of miRNAs than ALK7, as the inhibitor is less potent for ALK7 than for the other ALK members. Have the authors checked the expression levels of ALK4 and 5 in the hippocampus? Forth, in their rescue experiment presented in Figure 7 F2, are the levels for the three key miRNAs back to normal levels in trained animals?

We agree that these are important points. However, in our experiments we have not yet examined the levels of ALK4 and 5 nor the levels of miRNAs after translin reinstatement. We have discussed the reviewers points as future directions in the fifth paragraph of the Discussion section.

Finally, the Abstract is not doing justice to the outstanding data presented but contain more speculation than own data.

We have rewritten the Abstract as suggested.